# Evaluation of the Upper Airway in Class II Patients Undergoing Maxillary Setback and Counterclockwise Rotation in Orthognatic Surgery

**DOI:** 10.3390/cmtr18030039

**Published:** 2025-09-04

**Authors:** Flávio Fidêncio de Lima, Tayná Mendes Inácio De Carvalho, Bianca Pulino, Camila Cerantula, Mônica Grazieli Correa, Raphael Capelli Guerra

**Affiliations:** 1Department of Oral and Maxillofacial Surgery, Encore Clinic, São Paulo 04545-041, Brazil; 2Dental research Division, School of Dentistry, Universidade Paulista-UNIP, São Paulo 04026-002, Brazil; 3School of Dentistry of São José dos Campos, São Paulo State University (UNESP), São José dos Campos 12245-000, Brazil; 4Hospital Sírio-Libanês, Institute of Education and Research, São Paulo 01308-050, Brazil; 5Department of Oral and Maxillofacial Surgery, Leforte Hospital, Américas Health Network, São Paulo 01507-000, Brazil; 6Department of Oral and Maxillofacial Surgery, Hospital Israelita Albert Einstein, São Paulo 05652-900, Brazil; 7School of Dentistry of Araçatuba, São Paulo State University (UNESP), Araçatuba 16015-050, Brazil; 8Faculdade Israelita de Ciências de Saúde Albert Einstein, São Paulo 05653-120, Brazil

**Keywords:** orthognathic surgery, airway management, maxillary setback, counterclockwise rotation

## Abstract

Introduction: Maxillary setback in orthognathic surgery has been extensively discussed regarding its effects on bone healing and facial soft tissue profile; however, its impact on upper airway volume remains unclear. Objective: We evaluate the influence of maxillary setback combined with counterclockwise (CCW) rotation of the occlusal plane on upper airway dimensions. Methods: A retrospective observational case series was conducted with eight patients diagnosed with Class II malocclusion who underwent orthognathic surgery involving maxillary setback and CCW mandibular rotation. All procedures were performed by the same surgeon. Preoperative (T1) and 6-month postoperative (T2) facial CT scans were analyzed using Dolphin Imaging software11.7 to measure airway volume (VOL), surface area (SA), and linear distances D1, D2 and D3. Statistical analysis was performed using the Wilcoxon test with a 5% significance level. Results: Significant skeletal changes were observed, including 10.2 mm of mandibular advancement, 5.2 mm of hyoid advancement, and 4.1° of CCW rotation. Although increases in airway volume and surface area were noted, they did not reach statistical significance (*p* = 0.327 and *p* = 0.050, respectively), but suggesting a favorable trend toward airway adaptation. Conclusions: Maxillary setback combined with CCW rotation appears to safely correct Class II skeletal deformities without compromising upper airway space. These preliminary findings highlight the technique’s potential for both functional and aesthetic outcomes, warranting further long-term studies.

## 1. Introduction

The upper airway (UA) is a critical anatomical structure for respiratory function and is closely related to occlusion and craniofacial structure. Class II patients are generally narrower anteroposteriorly than in Class I individuals, associating skeletal deformity with potential respiratory function impairment [1,2,3].

Orthognathic surgery, particularly Le Fort I osteotomy with maxillary setback, is widely used to correct dentofacial deformities such as maxillary protrusion [4,5,6,7]. The maxillary setback, combined with CCW rotation of the occlusal plane, aims not only to improve facial aesthetics but also to offer functional benefits by altering the dimensions of the UA [7,8,9]. Manipulation of the occlusal plane during orthognathic surgery can significantly improve the UA volume, especially in the nasopharyngeal and oropharyngeal regions [10,11].

The combination of maxillary setback and CCW rotation of the occlusal plane provides evident aesthetic benefits for patients with maxillary protrusion, contributing to the normalization of the facial profile, particularly the nasolabial angle and upper lip projection [1,12,13]. Additionally, the occlusal plane rotation has favorable implications for the upper airway, such as widening and potential functional improvements, especially in cases of mandibular retrognathism and excessive maxillary vertical growth [6,14,15,16]. Movements like maxillary setback and CCW rotation can induce a significant increase in the UA, offering an additional functional benefit to the patients.

Therefore, the objective of this study is to evaluate the influence of maxillary setback and CCW rotation of the occlusal plane on the dimensions of the upper airway in Class II malocclusion patients undergoing orthognathic surgery. Based on the literature evidence suggesting a relationship between craniofacial structural manipulation and airway changes, this study aims to deepen the understanding of these interventions’ effects on respiratory function and aesthetic outcomes [10,17,18].

## 2. Materials and Methods

### 2.1. Study Design

This study was approved by the Ethics Committee (CAAE: 82049424.5.0000.0090). It is a retrospective, longitudinal, and observational case series conducted at the private Encore Clinic. The surgical procedures were performed at Hospital Edmundo Vasconcelos, located in São Paulo, São Paulo (SP, Brazil). The study followed the CARE Statement guidelines and checklist.

Eight patients diagnosed with type II dentofacial deformity and/or long face (Figure 1), who underwent orthognathic surgery involving maxillary setback or vertical impaction with CCW rotation of the occlusal plane, were included. All procedures were carried out by the same surgical team, following a standardized protocol, between February 2021 and January 2023.

Patients eligible for inclusion were those aged 18 years or older, with an SNA angle greater than 82 degrees, diagnosed with type II dentofacial deformity and long face, available for at least 6 months of follow-up, and able to speak, read, and understand the Portuguese language. Exclusion criteria comprised pregnancy, age under 18 years, facial skin infections, a history of minimally invasive treatment for temporomandibular disorders within the last 6 months, severe psychiatric illness or neurological disorders, medical contraindications to treatment or to radiographic/tomographic examinations, ongoing radiotherapy in the head and neck region, inability or unwillingness to provide informed consent, language barriers, or neuropsychomotor developmental delays.

### 2.2. Surgical Technique

All patients in this study underwent bimaxillary orthognathic surgery combined with genioplasty. A conventional Le Fort I osteotomy was performed, including pterygomaxillary separation using a curved chisel. Bony interferences were identified and removed to allow proper repositioning of the maxilla. When interference persisted after removal of the maxillary tuberosity, the pterygoid plate was ground or removed using bone forceps (Figure 2). Maxillary setback or vertical impaction was performed in combination with CCW rotation and internal rigid fixation. Fixation of the maxillary segment was achieved using two Lindorf miniplates at the piriform rim and two L-shaped miniplates at the zygomatic buttress, along with screws. For the mandibular procedure, a bilateral sagittal split ramus osteotomy (BSSRO) with a short lingual osteotomy was carried out. Each side was fixed using a 4-hole miniplate and screws, along with two or three bicortical positioning screws. Light elastic traction was applied for 3 to 4 weeks postoperatively. Mouth opening exercises began 3 weeks after surgery, aiming for an interincisal opening of over 40 mm within the following 3 weeks. Postoperative orthodontic treatment commenced 6 weeks after surgery. The final sample consisted of 8 patients, including 7 women and 1 man, with a mean age of 29.4 years.

### 2.3. CT Scan Acquisition and Image Analysis

CT scans were obtained using a Fan Beam System tomograph at two distinct time points: preoperatively, classified as T0, and at six months postoperatively, classified as T1. During imaging, patients were instructed to sit in an upright position with the head in its natural posture, looking straight ahead. Occlusion was maintained in centric relation. The helical beam CT scans were acquired with a slice thickness of 0.5 mm, under the following imaging parameters: 120 kVp, 150 mA, and an exposure time of 750 milliseconds. All data were saved in Digital Imaging and Communications in Medicine (DICOM) format.

Image processing and analysis were performed by a single calibrated operator (ELF) using Dolphin 3D Imaging software, version 11.7 Premium (Dolphin Imaging, Chatsworth, CA, USA). A single blinded examiner was responsible for all measurements. The software’s specific tools for airway analysis were used to assess the upper airway, including superior airway volume (VOL), surface area (A), and linear distances D1, D2, and D3 across the preoperative, immediate postoperative, and intermediate postoperative periods (Figure 3).

To standardize the airway analysis, anatomical limits were defined prior to measurement. The upper limit was determined by a line parallel to the Frankfort Horizontal Plane, extending from the apex of the axis vertebra (C2) beyond the airway boundaries. The anterior limit included the anterior oropharyngeal wall along with its overlying soft tissues. The lower limit corresponded to a line drawn parallel to the Frankfort Horizontal Plane between the lower border of the fourth cervical vertebra (C4) and the anterior oropharyngeal wall. The posterior limit was defined by a line running from the posterior margin of the dens of the axis (C2) to the lower border of the C4 vertebra, encompassing the posterior boundary of the airway space.

Once the region of interest was delineated, the ‘Add Seed Points’ tool was used to initiate the measurement process. Airway space detection sensitivity was standardized at 25%, after which the ‘Update Volume’ function was applied to calculate the volume of the airway segment. Finally, the ‘Measure Area and Angle’ tool was used to determine the cross-sectional area at the most constricted point, identified between two parallel planes placed at the uppermost and lowermost levels of the airway segment.

### 2.4. Statistical Analysis

Mean, median, standard deviation, minimum and maximum values, as well as interquartile ranges (first and third quartiles), were calculated for the variables under study. Comparisons between preoperative and postoperative measurements were performed using the Wilcoxon signed-rank test. Boxplots, scatter plots, and histograms were generated to visually represent the analyzed variables. A significance level of 5% was adopted. All analyses were conducted using SPSS for Windows, version 25.

## 3. Results

This study included eight patients with Class II dental deformity and/or a long face who underwent orthognathic surgery involving maxillary setback and CCW rotation of the occlusal plane. The sample consisted of seven women and one man, with an average age of 29.4 years (ranging from 20 to 45 years) (Table 1).

The results demonstrated significant skeletal changes, with a mean mandibular advancement of 10.20 mm (SD ± 4.17; *p* < 0.001) and a mean hyoid bone advancement of 5.20 mm (SD ± 2.79; *p* = 0.012), confirming the effectiveness of repositioning. The maxillary setback was moderate, with −1.25 mm at the posterior nasal spine (SD ± 2.67; *p* = 0.085) and −1.81 mm at the anterior nasal spine (SD ± 2.27; *p* = 0.042), indicating greater significance in the anterior repositioning. The CCW rotation of the occlusal plane was consistent, with a mean of 4.14° (SD ± 1.90; *p* = 0.003), reinforcing the precision of the surgical protocol. The vertical movement of the maxilla remained stable (0.12 mm; SD ± 2.38; *p* = 0.891), with no significant changes (Table 2).

In the airway analysis, the volume increased by 6.2% (*p* = 0.327), while the area showed a trend toward improvement (20%; *p* = 0.050), approaching statistical significance. Distance 2 demonstrated a significant increase (*p* = 0.058), suggesting a functional benefit (Table 3).

The distribution of measurements such as airway volume, surface area, and distances 1, 2, and 3, both preoperatively and six months after surgery, is shown in Figure 1, Figure 2, Figure 3, Figure 4 and Figure 5. Preoperative values are presented separately in Figure 6, Figure 7, Figure 8, Figure 9 and Figure 10.

## 4. Discussion

Orthognathic correction combining maxillary setback with CCW rotation of the occlusal plane aims to balance facial profile refinement with airway preservation in patients presenting Class II and long-face patterns. In this series, posterior maxillary movements at ANS and PNS were modest, whereas mandibular advancement reached approximately 10 mm and hyoid advancement approximately 5 mm, with a mean CCW rotation of about 4 degrees (Figure 11, Figure 12, Figure 13 and Figure 14). At six months postoperatively, airway volume exhibited a small and non-significant increase of roughly 6% (*p* = 0.327), while the cross-sectional area showed a borderline increase of 20% (*p* = 0.050). Distance 2 tended to enlarge (*p* = 0.058), Distance 1 showed a slight, non-significant decrease, and Distance 3 remained stable. These findings suggest an adaptive airway response rather than compromise when maxillary setback is intentionally limited and combined with CCW rotation and mandibular advancement.

The observed patterns are consistent with previous evidence indicating that isolated posterior repositioning of the maxilla may negatively affect the retropalatal segment, whereas CCW rotation and mandibular advancement are associated with enlargement of the pharyngeal airway and attenuation of potential narrowing. Mehra et al. reported increases in the pharyngeal airway space following CCW rotation of the maxillomandibular complex, while de Sousa Miranda et al. demonstrated three-dimensional expansion of the superior airway space in Class II patients after advancement with CCW rotation [15,16]. Conversely, studies addressing posterior impaction or setback have highlighted potential reductions in velopharyngeal and retropalatal dimensions, emphasizing the importance of compensatory surgical planning when maxillary setback is indicated [16]. Recent three-dimensional analyses following Le Fort I setback also describe nasopharyngeal changes, reinforcing that both the direction and magnitude of skeletal movement determine the airway response [16,17]. In the present study, the borderline increase in cross-sectional area and the tendency toward greater Distance 2 (oropharyngeal region) likely reflect the combined effect of mandibular and hyoid advancement offsetting any retropalatal susceptibility to narrowing.

The hyoid advancement documented here, measuring approximately 5 mm (*p* = 0.012), has clinical relevance, as anterior–inferior displacement of the hyoid complex may tension the suprahyoid musculature and promote hypopharyngeal patency, complementing the airway gains achieved through mandibular advancement [6,15,16,17]. Pereira et al. reported that such airway improvements may persist for up to four years, suggesting that part of the benefit results from skeletal and soft-tissue rebalancing with potential long-term stability [17]. Nevertheless, a recent systematic review of CBCT-based studies underscored the heterogeneity of segmentation protocols, anatomical landmarks, and follow-up intervals, factors that hinder direct inter-study comparisons and may attenuate measurable effects in smaller samples [18]. The standardized ROI definition used here (C2–C4) and consistent thresholding methodology ensured internal comparability, although the six-month observation period may be insufficient to capture late remodeling or relapse, as described in the literature [18].

From a surgical perspective, the protocol adopted in this cohort was designed to minimize retropalatal risk while achieving profile refinement. Limited setback with preservation of ANS projection, careful management of pterygomaxillary interferences, and stabilization at the piriform rim and zygomatic buttress were employed to avoid excessive posterior displacement of the nasomaxillary complex [10,12,19]. When indicated, adjunctive nasal airway procedures and subspinal or horseshoe osteotomy techniques described in the literature may further support nasal airflow and soft-tissue aesthetics without compromising airway safety [4,15]. Esteves et al. emphasized the role of occlusal-plane control in directing skeletal movements along favorable vectors, in line with our use of CCW rotation within a working range of 3 to 7 degrees. (Figure 11, Figure 12, Figure 13, Figure 14 and Figure 15) [9,15].

Two methodological points merit attention. First, although the change in airway volume was minimal, the near-significant increase in cross-sectional area suggests that minimal-area measurements may be more sensitive than global volume in detecting functionally relevant changes, an interpretation supported by previous three-dimensional and CBCT-based studies [15,16,17,18]. Second, the small sample size (n = 8) and predominance of female patients limit statistical power for detecting small-to-moderate effects, increasing the likelihood of type II error for findings with borderline significance (*p* = 0.050–0.058). Larger studies with stratification by sex, skeletal pattern severity, and precise movement vectors are needed to refine the dose–response relationship.

Functional correlation remains essential. While imaging in this series indicated airway stability or a slight positive trend at six months, future research should integrate three-dimensional analysis with validated sleep-quality instrument, such as the Pittsburgh Sleep Quality Index and its Brazilian Portuguese version, and, when possible, overnight monitoring to identify clinically meaningful outcomes in sleep-disordered breathing [18]. In a context of high aesthetic expectations and ethnic diversity, particularly relevant in Brazil, carefully planned non-conventional movements such as controlled maxillary setback can be performed safely when combined with CCW rotation and mandibular advancement, provided that both imaging and functional follow-up are systematically incorporated [14,18,19].

In summary, these findings support a surgical planning model in which limited maxillary setback combined with CCW rotation and mandibular/hyoid advancement achieves the desired aesthetic correction without measurable detriment to upper airway dimensions at six months, with trends suggesting improved cross-sectional patency (Figure 11, Figure 12, Figure 13, Figure 14 and Figure 15). These preliminary results align with previous literature and justify larger, long-term investigations to assess stability and to identify patient-specific predictors of airway response [14,20,21].

## 5. Conclusions

This study found that orthognathic surgery combining maxillary setback with counterclockwise rotation effectively corrected skeletal deformities in Class II and long-face patients. While mandibular and hyoid advancement occurred, airway volume changes after six months were not statistically significant (*p* > 0.05). Although successful for facial correction, the functional airway impact appears limited short-term. Further long-term studies with larger samples are needed to fully assess airway outcomes, despite the technique’s excellent aesthetic results.

## Data Availability

Data is contained within the article. The original contributions presented in this study are included in the article. Further inquiries can be directed to the corresponding authors.

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
