# Peer review of "Evaluation of the Upper Airway in Class II Patients Undergoing Maxillary Setback and Counterclockwise Rotation in Orthognatic Surgery"

_1943-3883, 2025, doi:10.3390/cmtr18030039_

Round 1

Reviewer 1 Report

Comments and Suggestions for Authors

Dear Authors,

In this article, the authors evaluated the effects of maxillary setback and counterclockwise rotation on the upper airway in 8 Class II cases.

The authors emphasized maxillary setback and counterclockwise rotation, but patients also underwent mandibular advancements and genioplasty. Therefore, it is difficult to determine the airway effect of maxillary setback and counterclockwise rotation alone.

Abstract

D1, D2, and D3 linear parameters should also be mentioned in the abstract section.

Introduction

Line 40 references are [9,16,17], but line 42 references are [1,2,3,19]. References should be checked in the whole manuscript.

Material and Methods

There seems to be a mistake in the descriptions of Figure 1. I think a-c is preoperative and b-d is postoperative.

Units are not written in the graphs, and chart descriptions can be written in more detail

Discussion

Why did the horseshoe Le Fort mention? Please explain the relationship of the study.

Study limitations should be mentioned. For example, the limited number of patients can be mentioned.

Author Response

Dear Editors and Reviewers,

We sincerely thank you for your valuable feedback and for the opportunity to improve our manuscript entitled:

“EVALUATION OF THE UPPER AIRWAY IN PATIENTS CLASS II UNDERGOING MAXILLARY SETBACK AND COUNTERCLOCKWISE ROTATION IN ORTHOGNATIC SURGERY” (manuscript ID: cmtr-3726502)

Below, we provide a point-by-point response to the reviewers’ comments, along with a summary of the corresponding changes made to the manuscript. All modifications have been highlighted in yellow throughout the revised version to facilitate your review.

Comment 1:

D1, D2, and D3 linear parameters should also be mentioned in the abstract section.

Response 1:

Thank you for review. The linear parameters D1, D2, and D3 have now been clearly described in the abstract to improve clarity and relevance.

Comment 2:

Line 40 references are [9,16,17], but line 42 references are [1,2,3,19]. References should be checked in the whole manuscript.

Response 2:

Thank you for your comment. All references have been reviewed and reorganized in numerical order for consistency. The reference list follows the ACS style, as described in the journal’s Instructions for Authors.

Comment 3:

There seems to be a mistake in the descriptions of Figure 1. I think a–c is preoperative and b–d is postoperative.

Response 3:

Thank you for your review. The description of Figure 1 has been corrected: “a–c” is now correctly labeled as preoperative and “b–d” as postoperative.

Comment 4:

Units are not written in the graphs, and chart descriptions can be written in more detail.

Response 4

Thank you for your comment. All graphs have been reviewed and reorganized The reference list follows the ACS style, as described in the journal’s

Comment 5:

Why did the horseshoe Le Fort mention? Please explain the relationship of the study.

Response 5 :

Dear Editor,

Thank you for your review and comments.

We referred to the study involving Le Fort osteotomy using the horseshoe technique as a basis, since it involves maxillary setback procedures. However, this study does not evaluate our main focus, which is the assessment of the airway. Both studies share a common interest in improving the aesthetics of the upper lip and nasal region.

Given that a large part of the literature highlights deficits not only in the aesthetics of this region but also in airway function, the aim of our study is to demonstrate that, when the diagnosis and surgical planning are appropriate, maxillary setback procedures may not compromise function. On the contrary, they can meet a cosmetic demand—especially in cases where there is an anteroposterior excess of the maxilla that persists even after all possible orthodontic movements have been performed, thus still requiring surgical setback.

Comment 6:

Study limitations should be mentioned. For example, the limited number of patients can be mentioned.

Response 6 :
Thank you for your review.

The fact that we included only eight patients with complete documentation who required the specific surgical movement—namely maxillary setback with counterclockwise rotation of the occlusal plane—does not invalidate the study due to sample size. We kindly ask the reviewers to consider that this is an extremely complex movement due to the anatomical structures involved in maxillary setback, and that it is generally not supported by the literature.

While many studies contraindicate pure maxillary setback, to our knowledge, no previous work has proposed an alternative such as this, which offers a valid option for Class II patients who often require maxillary setback and whose skeletal profile allows for counterclockwise rotation to achieve improved aesthetics.

Despite the limited sample size, our study was able to demonstrate, through statistical analysis, that in general there was no loss of respiratory function—and in some cases, there was even an increase in upper airway volume. In summary, we achieved facial aesthetic improvement in patients who required maxillary setback, while maintaining or even enhancing the upper airway dimensions..

Reviewer 2 Report

Comments and Suggestions for Authors

Current study shows a valuable contribution to the literature focusing on the the maxillary set-back combined with the counter-clockwise rotation addressing the aesthetic and functional outcome. However, the discussion could be improved. The small sample size, and short follow-up were not discussed in detail, especially considering the results. Although the changes are not discussed focusing the airway, functionality and aesthetic due the maxillary set-back and the mandibular counter-clockwise advancement. Additionally, it would add valuable information, to discuss and compare the findings of other authors.  Summerizeing, the clinical indications of this protocol could be discussed in more detail. A stronger and more detailed discussion would make the paper more complete and useful for readers. Consider to add the PSQI to this study which would strongly improve its value.

Author Response

(The authors gave the same response as above.)

Round 2

Reviewer 1 Report

Comments and Suggestions for Authors

Thanks for your work. Revisions have been made appropriately.

Author Response

The changes and improvements were made according to the reviewer's suggestions and guidelines in the text (discussion_). Thank you for your considerations.

Reviewer 2 Report

Comments and Suggestions for Authors

Thank you for the revision. Unfortunately, the main points from my first review were not really addressed.

The discussion is still very superficial and does not go enough into depth. Also, the changes about airway, functionality and aesthetic with the maxillary set-back and mandibular counter-clockwise rotation are not discussed. There is also no real comparison with other authors, so it is difficult to see how these results fit in the literature. It would add sustainable information for less experienced orthognathic surgeons.   

Because these points are very important for the value of the paper and they were not improved, I recommend the revision of this manuscript.

Author Response

(The authors gave the same response as above.)
